# The Role of MicroRNAs in Progressive Supranuclear Palsy—A Systematic Review

**DOI:** 10.3390/ijms25158243

**Published:** 2024-07-28

**Authors:** Aleksandra Ćwiklińska, Grzegorz Procyk, Dariusz Koziorowski, Stanisław Szlufik

**Affiliations:** 1Department of Neurology, Faculty of Health Sciences, Medical University of Warsaw, 03-242 Warsaw, Poland; s079989@student.wum.edu.pl (A.Ć.); dariusz.koziorowski@wum.edu.pl (D.K.); 21st Chair and Department of Cardiology, Medical University of Warsaw, Banacha 1A, 02-097 Warsaw, Poland; grzegorz.procyk@wum.edu.pl; 3Doctoral School, Medical University of Warsaw, 02-091 Warsaw, Poland

**Keywords:** progressive supranuclear palsy, microRNAs, biomarkers, atypical parkinsonian disorders

## Abstract

Progressive supranuclear palsy (PSP) is a rare, neurodegenerative movement disorder. Together with multiple system atrophy (MSA), Dementia with Lewy bodies (DLB), and corticobasal degeneration (CBD), PSP forms a group of atypical parkinsonisms. The latest diagnostic criteria, published in 2017 by the Movement Disorders Society, classify PSP diagnosis into defined, probable, and possible categories based on clinical examination. However, no single test is specific and sensitive for this disease. Microribonucleic acids (miRNAs) are promising molecules, particularly in the case of diseases that lack appropriate diagnostic and treatment tools, which supports exploring their role in PSP. We aimed to systematically review the current knowledge about the role of miRNAs in PSP. This study was registered in the Open Science Framework Registry, and the protocol is available online. Primary original studies, both clinical and preclinical, written in English and assessing miRNAs in PSP were included. Systematic reviews, meta-analyses, reviews, case reports, letters to editors, commentaries, conference abstracts, guidelines/statements, expert opinions, preprints, and book chapters were excluded. The following five databases were searched: Embase, Medline Ultimate, PubMed, Scopus, and Web of Science. Each database was last searched on 18 June 2024. Eventually, nine original studies relevant to the discussed area were included. The risk of bias was not assessed. The selected research suggests that miRNAs may be considered promising biomarkers in PSP. However, the exact involvement of miRNAs in the pathogenesis of PSP is still to be determined. Several microRNAs were found to be dysregulated in patients with PSP. This applies to both brain tissue and fluids like cerebrospinal fluid CSF or blood. Several miRNAs were found that could potentially be helpful in differentiating among PSP patients, PD patients, and healthy individuals. Although some correlations and alterations have already been found, this field requires much more research. MicroRNAs are exciting and promising small molecules, and their investigation into many diseases, including PSP, may lead to significant discoveries.

## 1. Introduction

### 1.1. Progressive Nuclear Palsy Diagnostic Criteria

Progressive supranuclear palsy (PSP) is a rare, neurodegenerative movement disorder. Together with multiple system atrophy (MSA), Dementia with Lewy bodies (DLB), and corticobasal degeneration (CBD), PSP forms a group of atypical parkinsonisms [1]. Neurodegeneration is caused by intracellular amyloidogenic protein aggregation. Parkinson’s Disease (PD), MSA, and DLB are referred to as synucleinopathies resulting from abnormal deposition of α-synuclein in brain cells, while PSP and CBD are referred to as tauopathies resulting from abnormal deposition of tau protein [2]. The first description of PSP was published in 1964. Even after 60 years, the diagnosis of this condition is still primarily based on clinical examination, with definitive confirmation achievable only through neuropathological examination of brain tissue, during which the aggregation of tau protein can be detected [3]. PSP is often underdiagnosed and frequently misdiagnosed as PD, especially in the early stages of the disease [4]. Despite some overlapping symptoms, PSP is characterized by distinct syndromes essential to distinguishing it from PD [5]. The latest diagnostic criteria, published in 2017 by the Movement Disorders Society, classify PSP diagnosis into defined, probable, and possible categories based on clinical examination [6]. PSP has many subtypes that differ in their clinical presentation, especially in the early stages [7]. The most common is PSP-Richardson syndrome, with vertical ocular motor dysfunction and the early beginning of postural instability [8]. Others include PSP-OM with an initial presentation in ocular motor dysfunction, PSP-P with a very similar clinical presentation to PD, PSP-PI with an early onset of postural instability, PSP-F with initial symptoms such as frontotemporal dementia, PSP-CBS with corticobasal syndrome, PSP-PGF with progressive gait freezing, and PSP-SL with an early manifestation of speech/language disorders [3,6,9,10,11,12]. The criteria primarily concern reaching high specificity and sensitivity in the diagnosis of PSP in general, and then also describe criteria for specific subtypes. The core diagnostic criteria include the most characteristic clinical symptoms of the disease including the following: ocular motor dysfunction, postural instability, akinesia, and cognitive dysfunction. The supportive features include levodopa resistance, a characteristic of atypical parkinsonism, hypokinesia, spastic dysarthria, dysphagia, and photophobia. Additionally, imaging tests can reveal midbrain atrophy [7]. In PSP, the grey matter of the brain is particularly affected. Studies describe grey matter reductions in subcortical and cortical areas in the frontal motor cortices, in the medial and lateral frontal cortices, and in the insula, striatum, thalamus, anterior cerebellum, and midbrain [13]. Degeneration of the midbrain can be found with MRI, which can help make the diagnosis [14].

### 1.2. Epidemiology

Diagnostic difficulties and low disease awareness make it difficult to create high-quality epidemiological data. The prevalence of this disease is approximately 7 per 100,000 people and increases with age [15]. Additionally, the average age of diagnosis is 72 years old with no gender predominance [16]. Because of the non-specific onset of the disease, diagnosis is usually delayed by 3–4 years from the onset of the first symptoms [6]. Most often, PSP is misdiagnosed as PD, Alzheimer’s Disease (AD), cognitive disorders, or depression [15]. As a fatal, incurable condition, the median survival of PSP patients is estimated at 5–8 years and depends on the phenotype of the disease [17,18]. However, survival data may vary because of differences in the definition of survival, either from symptom onset to death or from date of diagnosis to death [19]. The only confirmed risk factor for the disease is age, while some environmental factors, such as exposure to metals or chemicals, stress, and hypertension, are described as additional potential risk factors [20].

### 1.3. Problems in Clinical Practice

Lack of knowledge about PSP among general practitioners leads to delays in diagnosis, which poses difficulties in precisely describing the course of the disease and potential premorbid or early-stage factors and biomarkers [21]. However, even if the disease is diagnosed early, no disease-modifying therapies are yet available [22]. Furthermore, symptomatic treatment in most cases, especially in advanced stages, is also insufficient. Among patients with PSP, the effect of using levodopa is weak; it slightly relieves symptoms and has no impact on disease duration. Treatment focuses on alleviating symptoms separately, e.g., toxinum botulinum is used for eyelid opening [23]. Deep brain stimulation (DBS), commonly used in PD, was tested in PSP patients and showed no benefits [24]. In addition, some clinical trials tested tau-targeted therapy, but all were in the early stages [25]. PSP-specific biomarkers and targeted, effective treatment need to be further explored.

### 1.4. MicroRNAs as Novel Biomarkers

Microribonucleic acids (miRNAs, miRs) are small non-coding RNA molecules consisting of about 22 nucleotides [26]. They play an essential role in the regulation of gene expression. They can bind to an mRNA molecule, inhibiting mRNA translation or even leading to mRNA degradation [27]. Since their discovery in 1993 [28], they have been eagerly investigated in various conditions. MiRNA expression levels are altered in multiple diseases, including cardiovascular diseases, e.g., myocarditis [29,30,31] or aortic stenosis [32], Duchenne Muscular Dystrophy [33], and, more importantly, also neurodegenerative diseases like Alzheimer’s Disease, Parkinson’s Disease, and triplet repeat disorders [34]. MiRNAs can be found in solid tissues and fluids, e.g., blood, cerebrospinal fluid, and saliva [35]. This makes them promising potential biomarkers that could serve as diagnostic tools, particularly useful in conditions for which diagnosis is made in advanced stages of the disease, such as in PSP. MicroRNAs were also reported to help distinguish among the severities of various diseases [36,37]. Moreover, they can potentially be used to identify good- and poor-responders before treatment implementation [38]. Whether alterations in miRNA levels are causative of the investigated diseases or are only the result of a given condition must be determined for each disease individually. If such a cause-and-effect relationship is established, it opens the door for novel therapies based on synthetic oligonucleotides [39]. MicroRNAs are promising molecules, particularly in the case of diseases that lack appropriate diagnostic and treatment tools, which supports exploring their role in PSP.

We aimed to systematically review the current knowledge about the role of miRNAs in PSP. We aimed to answer the following questions: (1) Can miRNAs differentiate patients with PSP from healthy people? (2) Can miRNAs differentiate patients with PSP from patients with other forms of atypical parkinsonism? (3) Can miRNAs differentiate patients with PSP from patients with Parkinson’s disease? (4) Are miRNA levels altered in an animal model of progressive supranuclear palsy?

## 2. Methods

This systematic review was conducted according to the PRISMA 2020 Statement [40]. This study was registered in the Open Science Framework Registry, and the is protocol available online (DOI: https://doi.org/10.17605/OSF.IO/YSBQV).

Primary original studies, both clinical and preclinical, written in English and assessing miRNAs in PSP were included. Systematic reviews, meta-analyses, reviews, case reports, letters to editors, commentaries, conference abstracts, guidelines/statements, expert opinions, preprints, and book chapters were excluded. The following five databases were searched: Embase, Medline Ultimate, PubMed, Scopus, and Web of Science, by the following query: “(“miRNA” OR “microRNA”) AND (“PSP” OR “progressive supranuclear palsy” OR “progresive supranuclear palsy” OR “progressive supranucleal palsy” OR “progressive supranuclear palssy” OR “Steele–Richardson–Olszewski syndrome”)”, which yielded a total of 262 records. Each database was last searched on 18 June 2024. For each screening stage, we used 2 screeners working together (not independently). Final decisions in arguable cases were reached by a consensus between the 2 screeners. Data from each included work were extracted by a single extractor independently. In the case of any doubts, the issue was discussed with the other extractor.

The following data were extracted from each included paper: study characteristics: sample sizes, methodology used, year of publication, changes in miRNA levels among compared groups, and results of receiver operating characteristic (ROC) analysis (area under the ROC curve [AUC] with 95% CI). The risk of bias was not assessed.

After removing 121 duplicates, the remaining 141 records were screened by title and abstract. This yielded 16 records that met the inclusion/exclusion criteria. All 16 studies were retrieved in complete form. The complete data reports were evaluated for eligibility, excluding 7 studies as follows: 6 because of paper type (only abstract/poster as a post-conference material) and 1 because of inappropriateness. Eventually, 9 original studies relevant to the discussed area were included (Figure 1). The small number of included research papers allowed us to discuss and summarize all the existing knowledge in this field thoroughly.

We divided these studies into the following parts: (i) miRNA changes in a PSP animal model, (ii) miRNA changes in human brain tissues of PSP patients, and (iii) miRNA changes in the body fluids (CSF, blood/serum/plasma) of PSP patients.

## 3. Results and Discussion

### 3.1. miRNA Changes in a Progressive Supranuclear Palsy Animal Model

Lauretti et al. investigated miRNA changes in the mice model of tauopathy, which also encompasses PSP. They used wild-type C57BL/6 and hTau mice, which do not express endogenous mice tau but express all six isoforms of human tau protein. First, they compared the expression levels of several miRNAs in brain tissues (cortex, hippocampus, cerebellum) of hTau mice to their expression levels in wild-type mice at different time points. They found that miR-22-3p, miR-132-3p, miR-146a-5p, and miR-455-5p in the hippocampus and miR-132-3p and miR-146a-5p in the cerebellum were increased when assessed at the age of 12 months. At six months, miR-132-3p and miR-146a-5p in the cerebellum were also increased, but in the hippocampus, only miR-146a-5p was increased. At three months, there was no difference between hTau and wild-type mice. In addition, the authors assessed the time-related changes in hTau and wild-type mice. Exciting patterns were found in the cortex of wild-type mice; mir-22-3p, miR-132-3p, and miR-146a-5p increased over time, while no change was seen in hTau mice. In contrast, in the hippocampus of hTau mice, miR-132-3p and miR-146a-5p increased over time, while no change was found in wild-type mice. In the cerebellum of both hTau and wild-type mice, miR-22-3p decreased over time, while miR-146a decreased only in wild-type mice [41]. The studies discussed in this subsection with additional data are summarized in Table 1.

### 3.2. miRNA Changes in Human Brain Tissues of Progressive Supranuclear Palsy Patients

Smith et al. examined the following five miRNAs: miR-9, miR-124, miR-132, miR-137, and miR-153. MiRNA levels were measured in brain samples from PSP patients and healthy controls. Specifically, in the temporal, parietal, and prefrontal lobes. MiR-9, miR-124, miR-132, miR-137, and miR-153 increase the 3R-tau level, while miR-9, miR-132, and miR-137 decrease 4R-tau. The study confirmed that the 4R:3R-tau ratio was elevated in PSP patients compared with the healthy control group. MiR-132 expression was significantly decreased in the temporal lobes of PSP patients compared with healthy controls, and there were no differences for the other miRNAs. Moreover, the correlation between miR-132 levels and polypyrimidine tract binding protein 2 (PTB2) was examined. The results showed that miR-132 targets PTB2, which is increased in PSP patients and may lead to aberrant splicing of tau exon 10, increasing the risk of tauopathy [42].

Tatura et al. investigated miRNA in PSP in two steps. First, a microarray was performed to find dysregulated miRNA in the frontal lobe tissue of PSP patients compared to healthy controls. The test showed that four miRNAs were significantly dysregulated. MiR-147a and miR-518e were significantly upregulated in the PSP patients’ forebrains, while miR-504 and miR525-3p were significantly decreased. Confirmation of the results was performed using quantitative reverse transcription-polymerase chain reaction (qRT-PCR), which showed significant changes only in the levels of miR-147a and miR-518e, with no significant changes in the other two. In addition, the expression of genes targeted by miR-147a—NF1, ACLY, and ALG12—and by miR-518e—CPEB1 and JAZF1—was also examined. The results showed repression of these genes in the frontal lobe tissue of PSP patients, which correlated with increased expression of miR-147a and miR-518e. The targeted genes play a role in various cellular functions, so researchers suggested that their downregulation may be associated with disease occurrence [43]. Both studies discussed in this subsection, along with additional data, are summarized in Table 2.

### 3.3. miRNA Changes in the Body Fluids (Cerebrospinal Fluid, Blood/Serum/Plasma) of Progressive Supranuclear Patients

Starhof et al. evaluated the differences in the expression of 46 various miRNAs between PD patients with two atypical parkinsonism diseases—MSA and PSP—and a healthy control group. They measured miRNA expression levels in CSF and plasma. The results of the CSF examination showed that miR-106b-5p could differentiate between PD and PSP effectively. Similarly, miR-218-5p measured in plasma separated PSP and PD with good efficacy. However, no similarities were found between the results obtained in plasma and CSF. Furthermore, the relationship between α-synuclein and miRNA levels was examined, but no significant correlation was found [44].

Manna et al. evaluated the expression levels of several miRNAs in cohorts of PSP and PD patients and healthy controls. They found that miR-22-3p and miR-425-5p were upregulated in PSP patients compared with healthy controls, but this difference appeared non-significant after appropriate adjustments. However, the most efficient miRNA profile (miR-425-5p, miR-21-3p, miR-223-5p, miR-22-3p, miR-29a-3p, miR-483-5p) presented an AUC of 0.90 in differentiating between PSP patients and healthy controls. Furthermore, the authors found that miR-425-5p, miR-21-3p, and miR-199a-5p decreased in PSP patients compared with PD patients, and the differences remained significant even after adjustments. ROC analysis of these three miRNAs showed an AUC of 0.86 in discriminating between PSP and PD patients. Additionally, the best miRNA profile (consisting of miR-21-3p, miR-199a-5p, miR-425-5p, miR-483-5p, miR-22-3p, and miR-29a-3p) had an AUC of 0.91 in this setting. The authors analyzed the potential pathways of dysregulated miRNAs, and they found that the pathways most involved were fatty acid biosynthesis, ECM-receptor interaction, fatty acid metabolism, and the Hippo signaling pathway [45].

Nonaka et al. undertook an interesting study that included PSP patients and age- and sex-matched controls. They assessed miRNA levels in CSF, which makes it potentially useful in a clinical setting. The authors used a microarray technique capable of determining 2632 various miRNAs, of which 1104 were detectable in samples from both groups. They found that 38 different miRNAs were increased, while miR-6840-5p was decreased in PSP patients compared with controls. Unfortunately, the authors did not validate their results using the qRT-PCR method. However, they selected the two most upregulated and the most downregulated miRNAs in early-stage PSP patients (which were defined as patients who underwent neurological evaluation within two years of symptom onset) and then compared the levels of these miRNAs among early-stage PSP, advanced-stage PSP, and controls. Upregulation of miR-204-3p and miR-873-3p and downregulation of miR-6840-5p were found in early-stage PSP patients compared with controls. Similarly, miR-204-3p and miR-873-3p were increased, and miR-6840-5p was increased in advanced-stage PSP patients compared with controls. None of these miRNAs had altered expression levels between early-stage and advanced-stage PSP patients [46].

Ramaswamy et al. conducted a multi-stage study investigating plasma miRNAs’ potential role in PSP diagnosis. First, they performed miRNA profiling and found 28 dysregulated miRNAs in PSP patients; 23 were upregulated, while five were downregulated compared with healthy age-matched controls. Then, they evaluated miR-19b-3p, miR-33a-5p, miR-130b-3p, miR-136-3p, and miR-210-3p using the qPCR method. As expected, all the above miRNAs were upregulated in PSP patients compared to controls, consistent with the profiling results. The authors also studied the utility of these miRNAs in PSP diagnosis with ROC analysis. They found that miR-19b-3p, miR-33a-5p, miR-130b-3p, miR-136-3p, and miR-210-3p had an AUC of 0.7059, 0.8578, 0.7778, 0.7882, and 0.7810, respectively. They also assessed the diagnostic value of the combination of all five miRNAs and calculated an AUC of 0.7817 with a specificity of 66.67% and a sensitivity of 72.41%. Surprisingly, this multi-miRNA panel presented a lower AUC than, e.g., miR-33a-5p alone, which suggests not incorporating some of these miRNAs into such a panel. The authors also investigated the predicted target genes of the evaluated miRNAs, and they found 48 different pathways involved, of which 12 were targeted by at least two different miRNAs. Listing all the genes potentially targeted by these miRNAs is out of the scope of this review [47].

Simoes et al. investigated various non-coding RNAs as potential diagnostic tools in PSP patients. In addition to miRNAs, which we discuss below, they also measured piwi-interacting RNAs (piRs) and transfer RNAs. The authors measured miRNA concentrations in both serum and CSF. Nevertheless, the miRNAs that presented enough amplification to be analyzed differed in these two fluids. The only repeated one was piR-31068, but no correlation was found in its expression level between serum and CSF. Nonetheless, miR-92a-3p and miR-626 were shown to be downregulated in the serum of PSP patients compared with healthy controls. On the other hand, let-7a-5p was found to be upregulated in the CSF of PSP patients. The authors searched for potential target genes of dysregulated miRNAs. For miR-92a-3p, they identified the extracellular matrix–receptor interaction and regulation of the actin cytoskeleton as potentially involved pathways, while for miR-626, the epidermal growth factor receptor pathway, estrogen signaling pathway, and the phosphatidylinositol signaling system were identified. The potential targets for let-7a-5p were involved in the following pathways: cell cycle, lysine degradation, hippo signaling pathway, oocyte meiosis, extracellular matrix–receptor interaction, adherens junctions, and thyroid hormone signaling pathways [48].

Pavelka et al. conducted a study that included PSP patients, PD patients, and healthy controls. They performed miRNA profiling in whole blood samples with microarrays. They evaluated the differences in expression levels among the studied groups, and the potential diagnostic utility was assessed with ROC analysis. It was found that when compared with healthy controls, PSP patients presented upregulation of nine miRNAs, namely, miR-2115-5p, miR-4270, miR-505-3p, miR-769-5p, miR-3065-3p, miR-4638-5p, miR-197-3p, let-7d-3p, and miR-1225-5p, and downregulation of seven miRNAs, specifically, miR-4762-3p, miR-7975, miR-1233-5p, miR-6085, miR-125a-3p, miR-4465, and miR-564. Interestingly, the authors found no differences between PSP and PD patients in miRNA expression when adjusted for sex and age. Analysis with ROC revealed that miRNA panels were helpful in differentiating between PSP patients and healthy controls. In differentiating PSP and PD patients, miRNAs did not show high utility. However, miRNAs were potentially more useful in distinguishing between early-stage PSP and early-stage PD patients. In the analysis of predicted target genes, the most prominently affected pathway was the “BioCarta natural killer (NK) cell pathway” [49]. All studies discussed in this subsection with additional data are summarized in Table 3.

## 4. Discussion

Multiple microRNAs were found to be dysregulated in patients with PSP. This applies to both brain tissue and fluids like CSF or blood. Nevertheless, in general, miRNAs found to be altered in PSP do not overlap among the studies. This can be explained by the intrinsic limitations of research on miRNAs. In most studies, the researchers first explore potentially altered miRNAs in a small cohort of patients using microarrays. Then, they choose several miRNAs whose concentrations are altered the most significantly and perform qRT-PCR validation of these miRNAs on a bigger cohort of patients. Potentially, all miRNAs could be assessed with qRT-PCR, but this would be unreasonable in terms of the costs and time needed to perform such a study. Therefore, miRNAs that enter the validation step may substantially differ among studies since the first step is most often performed on four to five patients, and the effect size of miRNA alteration may significantly differ among studies even if the direction is consistent. The potential solution to overcome this limitation is to include miRNAs known to be altered from previous studies in the validation step; however, it is not always performed.

Notably, the measurement methods differed significantly among the analyzed studies. Only a few studies used qRT-PCR as a validation tool, while the others utilized microarrays alone, making it challenging to draw unambiguous conclusions.

However, miR-132 is worth mentioning in this context since it was upregulated in the cerebellum and hippocampus of mice models and downregulated in the temporal lobe of PSP patients. The opposite alteration could have been caused by (1) different brain parts where miRNAs were measured, (2) the fact that it was a mice model of tauopathy and not of PSP per se, and (3) the possibility that the mice model does not represent PSP in humans ideally. Interestingly, miR-132 is also downregulated in the brain of patients suffering from Alzheimer’s Disease, which is also a tauopathy like PSP [50].

As presented throughout this review, many studies not only compared the expression levels between PSP patients and healthy controls or PD patients, but they also performed an ROC analysis, which found several miRNAs that are potentially helpful in differentiating among PSP patients, healthy individuals, and PD patients. Nevertheless, we did not find a study that compared PSP patients with patients suffering from other forms of atypical parkinsonism. Although the study by Starhof et al. [43] included patients with MSA, the authors did not compare miRNA levels between PSP and MSA patients. Therefore, we cannot appropriately answer the question (2) Can miRNAs differentiate patients with PSP from patients with other forms of atypical parkinsonism?

Furthermore, although there are several subtypes of PSP, only a few studies presented exact patient characterizations with the number of each subtype. However, even in those studies, there was no analysis comparing the miRNA levels among various PSP subtypes.

## 5. Conclusions and Future Perspectives

The discussed studies suggest that miRNAs may be considered promising biomarkers in PSP. However, the exact involvement of miRNAs in the pathogenesis of PSP is still to be determined. Nevertheless, the observed alterations in miRNA concentrations, if validated by further research, may lead to the creation of a multi-biomarker panel. Including more miRNAs in such a panel would be associated with higher sensitivity and specificity but, on the other hand, it would lead to higher costs. Therefore, cost-effectiveness analysis would be needed to create a precise and affordable panel for use in daily clinical practice.

Although some correlations and alterations have already been found, this field requires much more research. Nevertheless, there are some difficulties, e.g., the fact that this condition is rare. For future research, we recommend validating results with qRT-PCR. It would be of great value to perform a study that includes healthy individuals from whom blood samples are collected at baseline and then correlate the baseline miRNA expression levels with PSP occurrence. This would require a prominent follow-up period of about 20-40 years. However, this could be performed using biobanks like the U.K. Biobank. 

Another interesting research field is comparing various PSP subtypes. Finding molecular biomarkers could even lead to reclassification, which is currently based primarily on clinical presentation.

MicroRNAs are exciting and promising small molecules, and their investigation into many diseases, including PSP, may lead to significant discoveries. Figure 2 presents a comprehensive summary of the current knowledge about the role of miRNAs in PSP.

## 6. Limitations

We must disclose a few limitations of this systematic review. First, this study reviewed only miRNAs and no other non-coding RNAs. Nevertheless, such a narrowing of the research topic allowed us to discuss it exhaustively within the space expected of a review paper. Secondly, this systematic review included different study types. However, this can be explained by a very limited amount of research in the field. Last, we did not assess the risk of bias, which may limit drawing firm conclusions. However, this might not be pivotal to such a small number of studies available.

## Figures and Tables

**Figure 1 ijms-25-08243-f001:**
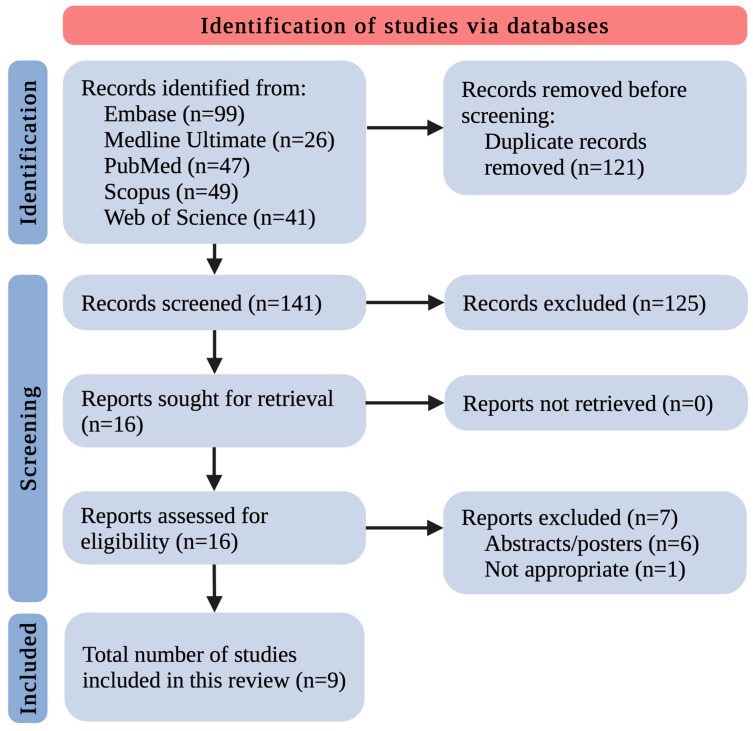
The flowchart for the selection process; n—number of studies.

**Figure 2 ijms-25-08243-f002:**
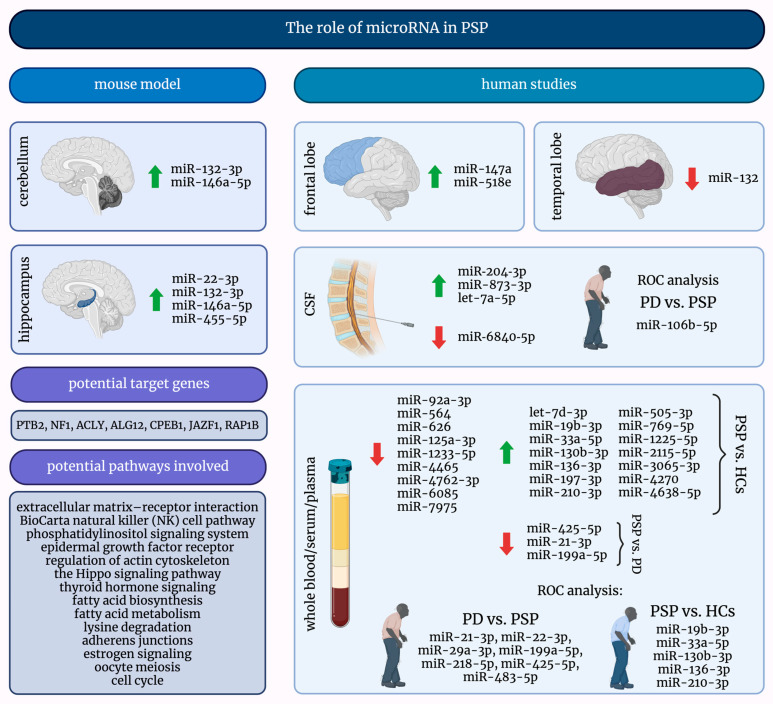
A graphical summary of the role of microRNAs in progressive supranuclear palsy. ↑—increased; ↓—decreased; HCs—healthy controls; miRNAs/miRs—micro-ribonucleic acids; PD—Parkinson’s Disease; PSP—progressive supranuclear palsy; ROC—receiver operating characteristic.

**Table 1 ijms-25-08243-t001:** Summary of recent studies regarding miRNA changes in a progressive supranuclear palsy animal model.

Ref.	Year	Population	Comparison	miRNA	Outcome	Methodology
[41]	2021	hTau mice (n = 3–8/group depending on the experiment)	wild-type C57BL/6J mice (n = 3–8/group depending on the experiment)	miR-22-3p, miR-132-3p, miR-146a-5p, miR-455-5p	12 months of age: ↑ miR-22-3p, miR-132-3p, miR-146a-5p, miR-455-5p in the hippocampus and ↑ miR-132-3p, miR-146a-5p in the the cerebellum of hTau mice 6 months of age: ↑ miR-146a-5p in the hippocampus and ↑ miR-132-3p, miR-146a-5p in the the cerebellum of hTau mice 3 months of age: no difference	miRNA in brain tissues (cortex, hippocampus, cerebellum) by qRT-PCR

↑—increased, miR/miRNA—microRNA, n—number of individuals, qRT-PCR—quantitative reverse transcription-polymerase chain reaction, ref.—reference, RNA—ribonucleic acid.

**Table 2 ijms-25-08243-t002:** Summary of recent studies regarding miRNA changes in human brain tissues of progressive supranuclear palsy patients.

Ref.	Year	Population	Comparison	miRNA	Outcome	Methodology
[42]	2011	8 PSP pts	8 HCs	miR-9, miR-124, miR-132, miR-137, miR-153	↓ miR-132 in the temporal lobes of PSP patients compared with HCs	miRNA in brain tissues by qRT-PCR
[43]	2016	20 PSP pts	20 HCs	miR-147, miR-518e, miR-504, miR-525-3p	↑ miR-147a and miR-518e in the frontal lobes of PSP patients compared with HCs	miRNA in frontal lobe tissue by qRT-PCR

↑—increased, ↓—decreased, HCs—healthy controls, miR/miRNA—microRNA, PD—Parkinson’s Disease, PSP—progressive supranuclear palsy, pts—patients, qRT-PCR—quantitative reverse transcription-polymerase chain reaction, ref.—reference, RNA—ribonucleic acid.

**Table 3 ijms-25-08243-t003:** Summary of recent studies regarding miRNA changes in body fluids (cerebrospinal fluid, blood/serum/plasma) of progressive supranuclear patients.

Ref.	Year	Population	Comparison	miRNA	Outcome	Methodology
[44]	2019	32 PSP pts	37 PD pts 29 MSA pts 23 HCS	46 various miRNAs	ROC analysis (PD vs. PSP) CSF: miR-106b-5p AUC 0.85 (95% Cl: 0.757–0.945) plasma: miR-218-5p AUC 0.71 (95% Cl: 0.594–0.826)	miRNA in CSF and plasma by qRT-PCR
[45]	2021	20 PSP pts	40 PD pts 33 HCs	miR-425-5p, miR-21-3p, miR-223-5p, miR-22-3p, let-7i-5p, miR-199a-5p, miR-29a-3p, miR-483-5p	↓ miR-425-5p, miR-21-3p, and miR-199a-5p in PSP pts compared with PD pts ROC analysis (PSP vs. PD) combination of miR-425-5p, miR-21-3p, and miR-199a-5p with AUC 0.86 (95% CI: 0.74-0.97) combination of miR-21-3p, miR-199a-5p, miR-425-5p, miR-483-5p, miR-22-3p, and miR-29a-3p with AUC 0.91 (95% CI: 0.82–1.00)	exosomal miRNA in serum by qRT-PCR
[46]	2022	11 PSP pts	8 age- and sex-matched controls	2632 various miRNAs	↑ 38 various miRNAs and ↓ miR-6840-5p in PSP pts compared with controls ↑ miR-204-3p and miR-873-3p and ↓ miR-6840-5p in early-stage PSP pts compared with controls ↑ miR-204-3p and miR-873-3p and ↓ miR-6840-5p in advanced-stage PSP pts compared with controls	miRNA in CSF by microarray analysis
[47]	2022	18 PSP pts	17 age-matched HCs	miR-19b-3p, miR-33a-5p, miR-130b-3p, miR-136-3p, miR-210-3p	↑ miR-19b-3p, miR-33a-5p, miR-130b-3p, miR-136-3p, miR-210-3p in PSP pts ROC analysis: miR-19b-3p AUC 0.7059 miR-33a-5p AUC 0.8578 miR-130b-3p AUC 0.7778 miR-136-3p AUC 0.7882 miR-210-3p AUC 0.7810 combination of all five miRNAs: AUC 0.7817 (95% CI: 0.7126–0.8508)	miRNA in plasma by qPCR
[48]	2022	31 PSP pts	20 age- and sex-matched HCs	let-7a-5p, let-7b-5p, let-7f-2-3p, miR-1-3p, miR-16-5p, miR-92a-3p, miR-148a-3p, miR-626, miR-3168	↓ miR-92a-3p, miR-626 in serum of PSP pts ↑ let-7a-5p in CSF of PSP pts	miRNA in CSF and serum by qRT-PCR
[49]	2024	35 PSP pts	367 PD pts 416 HCs	2549 various miRNAs	↑ miR-2115-5p, miR-4270, miR-505-3p, miR-769-5p, miR-3065-3p, miR-4638-5p, miR-197-3p, let-7d-3p, and miR-1225-5p and ↓ miR-4762-3p, miR-7975, miR-1233-5p, miR-6085, miR-125a-3p, miR-4465, and miR-564 in PSP pts compared with HCs no difference in miRNA expression between PSP and PD pts	miRNA in whole blood by microarray analysis

↑—increased, ↓—decreased, AUC—area under the ROC curve, CI—confidence interval, CSF—cerebrospinal fluid, HCs—healthy controls, miR/miRNA—microRNA, MSA—multiple system atrophy, qPCR—quantitative polymerase-chain-reaction, PD—Parkinson’s Disease, PSP—progressive supranuclear palsy, pts—patients, qRT-PCR—quantitative reverse transcription-polymerase chain reaction, ref.—reference, RNA—ribonucleic acid, ROC—receiver operating characteristic.

## Data Availability

All data will be made available from the authors upon request.

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
