# Peer review of "The Role of MicroRNAs in Progressive Supranuclear Palsy—A Systematic Review"

_ijms, 2024, doi:10.3390/ijms25158243_

Round 1

Reviewer 1 Report

Comments and Suggestions for Authors

Main criticisms and limitations of the study on MicroRNAs in Progressive Supranuclear Palsy (PSP) include:

1. The study notes that the small number of studies may restrict its generalizability and ability to draw clear conclusions. Please improve it.

2. Using different measuring methods like qRT-PCR and microarrays makes it hard to compare results and draw conclusions.

3. The study implies that MicroRNAs may be interesting indicators in PSP, although qRT-PCR is needed to confirm the findings.

4. A multi-biomarker panel using multiple MicroRNAs for PSP diagnosis may boost sensitivity and specificity but increase cost. Cost-effectiveness analysis is needed to assess the clinical utility of such a panel.

5. Given PSP's rarity, the report recommends greater investigation. Future studies should validate results with qRT-PCR and examine long-term connections between baseline MicroRNA expression and PSP.

6. The systematic review included several study types, which may affect evidence quality. Future bias assessments could improve study conclusions.

7. The study only considered microRNAs, not other non-coding RNAs. This allowed for a complete assessment, but exploring other non-coding RNAs could help explain PSP pathophysiology.

8. Author data access upon request may restrict study transparency and reproducibility. Open data could boost research credibility.

9. Future research on MicroRNAs in PSP may improve diagnostic and treatment methods for this neurodegenerative condition by addressing these suggestions and limitations.

The paper on microRNAs in Progressive Supranuclear Palsy (PSP) has some minor remarks and grammatical errors:

Minor Comments:

1. To ensure language consistency, define "miRNAs" at the start.

For those unfamiliar with abbreviations like "qRT-PCR" and "ROC", define them.

Use standard vocabulary and structure for references and citations throughout the work.

2. Mistakes in grammar:

Check sentence subject-verb agreement for clarity and accuracy.

Revise sentence structure for readability and flow. Check commas and semicolons for grammar and syntax.

The study can improve its clarity, coherence, and quality for Progressive Supranuclear Palsy and MicroRNA readers and researchers by resolving these minor criticisms and grammar problems.

Reviewer 2 Report

Comments and Suggestions for Authors

This is an interesting systematic review about role of micro-RNAs in PSP. The paper is generally well-written and has a good structure.

The introduction section gives the background of this review, describing the diagnostic criteria of PSP, its epidemiology, the related clinical problems in real-world setup and the micro-RNAs as novel biomarkers. The authors also mention the aim and the questions of the current review.

The methodology section is well written and the flow diagram also helps.

The findings of this review are well written in the results section and depicted in tables.

I think that the authors should add a separate discussion section, focusing on the main findings of the review and perhaps adding their critical view on the subject

The conclusions are well written and the authors have also mentioned some potential targets for future targets.

References are relative to the subject, adequate in number and relatively recent.

Reviewer 3 Report

Comments and Suggestions for Authors

This manuscript can be summarized as follows. Progressive supranuclear palsy (PSP) is a rare neurodegenerative movement disorder. The 2017 diagnostic criteria classify PSP into defined, probable, and possible categories based on clinical examination, but no single test is specific and sensitive for this disease. MicroRNAs (miRNAs) are promising molecules for PSP research. This study aimed to systematically review current knowledge about miRNAs in PSP. Five databases were searched, resulting in 9 original studies being included. The selected research suggests miRNAs may be promising biomarkers in PSP. Several miRNAs were found to be dysregulated in PSP patients' brain tissue, cerebrospinal fluid, and blood. Some miRNAs could potentially differentiate between PSP patients, Parkinson's disease patients, and healthy individuals. However, the exact involvement of miRNAs in PSP pathogenesis remains undetermined. While some correlations and alterations have been found, this field requires much more research. The investigation of miRNAs in PSP and other diseases may lead to significant discoveries. I have no grammar, misspellings or omissions in my manuscript.

1)     There is a need to add the limitations of miRNAs if they are not written. In the manuscript so far, there is no match of miRNAs in brain, spinal fluid, or plasma when distinguishing between PSP and healthy subjects. Similar results have been observed for miRNAs in neurodegenerative diseases. Please clarify the above. Ann Indian Acad Neurol. 2021 Jan-Feb;24(1):56-62. doi: 10.4103/aian.AIAN_611_20. Epub 2021 Jan 19.

2)     Subtypes of PSP and Prognosis: A Retrospective Analysis When categorized into subtypes, out of the 334 patients with PSP, PSP-RS predominated (72%), followed by PSP-parkinsonism (PSP-P) (13.5%), PSP-corticobasal syndrome (PSP-CBS) (5.1%), PSP-frontal (PSP-F) (4.2%), PSP-progressive gait freezing (PSP-PGF) (4.2%), PSP-postural instability (PSP-PI) (0.6%), and PSP-speech/language (PSP-SL) (0.3%). PSP-P reaches the milestones of wheelchair dependency, unintelligible speech, and dysphagia later than other subtypes. This study reviews 338 only miRNAs and no other non-coding RNAs. Nevertheless, such a narrowing of the Int. J. Mol. Sci. 2023, 24, x FOR PEER REVIEW 11 of 13 research topic allowed us to discuss it exhaustively within the space expected of a review manuscript. Differentiating PSP-parkinsonism (PSP-P) from PD is important, Nonaka, W.; Takata, T.; Iwama, H.; Komatsubara, S.; Kobara, H.; Kamada, M.; Deguchi, K.; Touge, T.; Miyamoto , O.; Naka-451 mura, T.; et al. A cerebrospinal fluid microRNA analysis: Progressive supranuclear palsy. does not differentiate between PSP-P and PD. It does not prove 2) Can miRNAs differentiate patients with PSP from pa-118 tients wtih other forms of atypical parkinsonism? You should be more precise on this point as well.

Reviewer 4 Report

Comments and Suggestions for Authors

Ćwiklińska and co-workers investigated the roles of miRNAs in a rare, neurodegenerative movement disorder, progressive supranuclear palsy (PSP). The manuscript discusses 9 research article in detail, and it is definitely of interest.

There are some questions/issues to consider:

1.       The introduction on the diagnostic criteria and epidemiology gives a detailed account concerning PSP. Which brain area/areas are affected in the patients?

2.       The authors aimed to answer 4 questions (line 116-121). They indeed have answered most of them, except question 2: “Can miRNAs differentiate patients with PSP from patients with other forms of atypical parkinsonism?”

3.       line 182-183: “MiR-132 expression was significantly decreased in the temporal lobes of PSP patients compared to healthy controls, and there were no differences for other miRNAs. “ But in table 2, there is an increase for miR-132.

4.       There is an opposite alteration in human and mice concerning the miR-132 level. Also, there is almost no common miRNA found in the various studies. What can be the reason for this? The authors mentioned in the introduction that there are many subtypes of PSP. Are the studies considered these subtypes?

5.       Although PSP belongs to atypical parkinsonism, it is a tauopathy. Might be there common miRNAs in PSP and Alzheimer’s disease?

Round 2

Reviewer 1 Report

Comments and Suggestions for Authors

The author improve the MS.